# Cigarette taxation and neonatal and infant mortality: A longitudinal analysis of 159 countries

**Márta K. Radó**[1,2☯], **Anthony A. Laverty**[3☯*], **Thomas Hone**[3], **Kiara Chang**[3], **Mohammed Jawad**[3], **Christopher Millett**[3], **Jasper V. Been**[1,2,4], **Filippos T. Filippidis**[3]

**1** Division of Neonatology, Department of Paediatrics, Erasmus MC Sophia Children's Hospital, University Medical Centre Rotterdam, Rotterdam, The Netherlands, **2** Department of Public Health, Erasmus MC, University Medical Centre Rotterdam, Rotterdam, The Netherlands, **3** Public Health Policy Evaluation Unit, School of Public Health, Imperial College London, London, United Kingdom, **4** Asthma UK Centre for Applied Research, Usher Institute, The University of Edinburgh, Edinburgh, United Kingdom

☯ These authors contributed equally to this work.
* a.laverty@imperial.ac.uk

**Data Availability Statement:** Data were collected from the World Bank Database, World Health Organization, and International Cigarette Consumption Database at

## Abstract

Previous studies on the associations between cigarette taxes and infant survival have all been in high-income countries and did not examine the relative benefits of different taxation levels and structures. We evaluated longitudinal associations of cigarette taxes with neonatal and infant mortality globally. We applied country-level panel regressions using 2008–2018 annual mortality and biennial WHO tobacco taxation data. Complete data was available for 159 countries. Outcomes were neonatal and infant mortality. We conducted analyses by type of taxes (i.e. specific cigarette taxes, ad valorem taxes, and other taxes, import duties and VAT) and the income group classification of countries. Covariates included scores for other WHO recommended tobacco control policies, socioeconomic, health-care, and air quality measures. Secondary analyses investigated the associations between cigarette tax and cigarette consumption. We found that a 10 percentage-point increase in total cigarette tax as a percentage of the retail price was associated with a 2.6% (95% Confidence Interval [CI]: 1.9% to 3.2%) decrease in neonatal mortality and a 1.9% (95% CI: 1.3% to 2.6%) decrease in infant mortality globally. Estimates were similar for both excise and ad valorem taxes. We estimated that 231,220 (95% CI: 152,658 to 307,655) infant deaths could have been averted in 2018 if all countries had total cigarette tax at least 75%. 99.2% of these averted deaths would have been in low- and middle-income countries (LMICs). The secondary analysis supported causal interpretation of results by finding that a 10 percentage-point increase in taxes was associated with a reduction of 94.6 (95% CI: 32.7 to 156.5) in annual cigarette consumption per capita. Although causal inference is precarious due to the quasi-experimental design, we used a robust analytical approach and focused on within-country changes. Limitations include an inability to include data on roll-your-own tobacco, other forms of tobacco use, and reliance on taxation data only for the cigarette brands most sold in each country. In line with limited existing evidence conducted in HICs, we found that raising taxes on tobacco was associated with a reduction in neonatal and infant mortality globally. Implementing recommended levels of taxation in LMICs should be a priority since

https://doi.org/10.1136/bmj.l2231, https://www.who.int/data/maternal-newborn-child-adolescent-ageing, and https://data.worldbank.org/.

**Funding:** The authors received no specific funding for this work.

**Competing interests:** The authors have declared that no competing interests exist.

this is where the lowest levels of taxation and the largest potential infant mortality benefits exist.

## Introduction

Tobacco use has substantial adverse impacts on child health globally. Exposure to second hand smoke (SHS) is estimated to kill 56,000 children under ten years of age each year around the globe primarily through exposure at home [1,2]. Smoking during pregnancy or the exposure of pregnant women to SHS is the second main pathway through which smoking increases risks for adverse early life health outcomes, including neonatal and infant mortality [3–5]. Tobacco use can also be the cause of catastrophic health expenditure for families, which may also indirectly affect children's health [6]. Therefore tobacco control has been considered a key strategy in the United Nations' Sustainable Development Goals (SDGs) 3.2.1 and 3.2.2, i.e. devoted to improving under-five and neonatal mortality rates [7].

The World Health Organization (WHO) recommends six measures to reduce tobacco use as part of the MPOWER strategy (i.e. Monitor tobacco use; Protect people from SHS; Offer help to quit tobacco use; Warn about the dangers of tobacco; Enforce bans on tobacco advertising, promotion and sponsorship; and Raise taxes). Raising taxation on tobacco has been shown to be the most effective measure with well-documented health benefits among adults, especially among low-income populations [8,9]. Studies examining the impact of cigarette taxation on child health outcomes are fewer than among adults [10]. While existing studies show positive impacts of increased tobacco taxation on preterm birth, infant mortality and asthma exacerbations, this research has been conducted only in high-income countries (HICs) [11–15]. These results however may not be generalisable to low- and middle-income countries (LMICs) where high background air pollution, low awareness of tobacco-related harm, poor economic conditions, and high influence of the tobacco industry might suppress the positive effects of raising taxes [16–20]. The generalisability of results is also hindered by the fact that the prevalence of smoking among women in their reproductive years (including also pregnant women) is significantly lower in LMICs than in HICs [16,21]. The dearth of research globally, especially in LMICs with the largest burden of child mortality, may constrain further progress in tobacco control. This is important as only 14% of the global population live in countries which achieve the WHO recommended level of taxation of at least 75% of the retail price [8].

This study aims to assess associations between cigarette tax and neonatal and infant mortality (i.e. deaths within 28 days and one year after birth, respectively) globally as well as separately in HICs and LMICs. We additionally examine how different tax regimes (specific excise, ad valorem, Value Added Tax (VAT)/ import duties/ other taxes) impact these associations and estimate the forgone global child health benefits of countries not achieving the WHO recommended levels of taxation on tobacco.

## Methods and materials

We conducted panel regression analyses using country-level data to model trends in cigarette taxation levels and neonatal and infant mortality and their associations as well as associations between cigarette taxation and cigarette consumption.

### Data

Annual or biennial country-level data on a range of variables were obtained between 2008 and 2018 for all countries with available data. Annual data on taxation variables and MPOWER

covariates were obtained from biennial data through linear interpolation using available data from previous and subsequent years. Data on outcomes were available for 194 countries (data were not available for the Occupied Palestinian Territories). After accounting for the presence of data on relevant covariates (detailed below), final models were run on 159 countries (S1 Text), with a lack of data on female primary education accounting for most of these exclusions. No data were imputed for outcome variables. Table 1 provides an overview of the outcomes, exposure variables, and covariates. Data were collected from the World Bank Database, World Health Organization, and International Cigarette Consumption Database [22–24].

## Outcomes

Data on neonatal and infant mortality were extracted from the WHO's 'Maternal, newborn, child and adolescent health and ageing' data portal [23]. Neonatal mortality was defined as 'the number of neonates who die within 28 days after birth per 1,000 livebirths' and infant mortality as 'the number of deaths among infants younger than one year of age per 1,000 livebirths'. We used annual data for each year from 2008 to 2018. Neonatal and infant mortality were not normally distributed across countries and over time; therefore, values were log-transformed. Data on annual population-weighted cigarette consumption per adult (in sticks) for 71 countries were taken from the International Cigarette Consumption Database (ICCD) [22,25]. The database includes data up to 2015.

## Exposure variables

Data on cigarette taxation were extracted from the WHO reports on the global tobacco epidemic between 2009 and 2019 [8,26–30]. These reports have been published biennially since 2009 and include data on cigarette taxation in the previous year, hence we obtained taxation data from 2008 to 2018. Data include tax as a percentage of the retail price of a pack of cigarettes of the most sold brand in each country. In some of the analyses, we considered the total cigarette tax as a categorical variable (<25%; 25–49.9%; 50–74.9%; ≥75% of the price). Data are available by type of tax, i.e. specific excise tax; ad valorem tax; and other taxes (VAT, import duties and other taxes), as well as the sum of all taxes, which we refer to as total cigarette tax.

## Covariates

Covariates were selected based on their previously demonstrated impact on the outcomes and exposure variables [13,31]. All models were adjusted for Gross Domestic Product (GDP) per capita, fertility rate, and MPOWER tobacco control measures (excluding 'Raise taxes', as we analysed detailed taxation data, and scores for 'Monitoring of tobacco use', as this measure is not expected to have an impact on the outcomes). MPOWER scores come from relevant WHO reports which give a score from zero to four on each domain and we modelled these continuously. Additionally, models with neonatal and infant mortality as outcomes were further adjusted for health expenditure per capita, % rural population, % access to drinking water, % access to clean cooking, and % female primary education completion rate, whereas cigarette consumption models were further adjusted for total primary education completion rate. MPOWER scores were obtained from the WHO reports and other covariates were extracted from the World Bank database (Table 1).

## Statistical analyses

We ran multiple linear panel regression models utilising a fixed-effects specification. Fixed-effects panel regression accounts for the hierarchical nature of the data (year observations

**Table 1. Description of the included variables.**

| Type of variable | Variables | Data source | Definition | Period | Frequency |
|---|---|---|---|---|---|
| Outcomes | Neonatal mortality | WHO | The number of neonates who die within 28 days after birth per 1,000 livebirths | 2008–2018 | Annual |
| | Infant mortality | WHO | The number of deaths among infants younger than one year of age per 1,000 livebirths | 2008–2018 | Annual |
| | Cigarette consumption | International Cigarette Consumption Database | Estimated cigarette consumption per capita data from tobacco sales data | 2008–2015 | Annual |
| Exposure variables | Total tax | WHO | Total tax as % of the price of the most sold brand (per 10%) | 2008–2018 | Biennial |
| | Specific tax | WHO | Specific excise tax (i.e. fixed amount per cigarette/ weight of tobacco) as % of the price of the most sold brand (per 10%) | 2008–2018 | Biennial |
| | Ad valorem | WHO | Ad valorem excise tax (i.e. a percentage of the factory price/ retail price) as % of the price of the most sold brand (per 10%) | 2008–2018 | Biennial |
| | Import duties, VAT, and other taxes | WHO | VAT/Sales (i.e. general tax on consumption), import duties (i.e. a tax on imported goods that are destined for domestic consumption), and other taxes (i.e. differently named taxes) as % of the price of the most sold brand (per 10%) | 2008–2018 | Biennial |
| Covariates for neonatal/ infant mortality and cigarette consumption | Protect people | WHO | Five-point scale based on WHO evaluation about "Protecting people from tobacco smoke" | 2008–2018 | Biennial |
| | Offer help to quit | WHO | Five-point scale based on WHO evaluation about "Offering help to quit tobacco use" | 2008–2018 | Biennial |
| | Warning about dangers: health warnings | WHO | Five-point scale based on WHO evaluation about "Health warnings" | 2008–2018 | Biennial |
| | Warning about dangers: mass media campaigns | WHO | Five-point scale based on WHO evaluation about "Mass media" | 2008–2018 | Biennial |
| | Enforce bans | WHO | Five-point scale based on WHO evaluation about "Enforcing bans on tobacco advertising, promotion and sponsorship" | 2008–2018 | Biennial |
| | GDP | World Bank | Gross domestic product per capita (PPP, per 1000) | 2008–2018 | Annual |
| | Fertility rate | World Bank | The average number of children born to a woman (given women survive the childbearing age and fertility is in line with age-specific fertility rates of the specified year) | 2008–2018 | Annual |
| Additional covariates for neonatal/ infant mortality | Rural population | World Bank | The proportion of the population living in rural areas as defined by national statistical offices (per 10%) | 2008–2018 | Annual |
| | Drinking water | World Bank | The proportion of the population with access to basic drinking water (i.e. collection time < 30 minutes) (per 10%) | 2008–2017 | Annual |
| | Health expenditure | World Bank | Current health expenditure per capita expressed in international dollars (PPP, per 1000) | 2008–2017 | Annual |
| | Female primary education completion rate | World Bank | The ratio of the number of new female entrants in the last grade of primary education (regardless of age) and the number of females at the entrance age for the last grade of primary education (per 10%) | 2008–2018 | Annual |
| | Clean cooking | World Bank | The proportion of the population with access to clean fuels and technologies for cooking (per 10%) | 2008–2018 | Annual |
| Additional covariates for cigarette consumption | Total primary education completion rate | World Bank | The ratio of the number of new entrants in the last grade of primary education (regardless of age) and the population size at the entrance age for the last grade of primary education | 2008–2018 | Annual |

Abbreviations: WHO = World Health Organization, VAT = value-added tax; GDP = Gross domestic product; PPP = Purchasing power parity.

clustered in countries) and also adjust for country-level fixed effects. Hence the models estimate associated changes within countries (i.e. the associations between changes in cigarette tax and change in outcomes) accounting for time-invariant country-level factors. The Hausman specification test indicated that the random-effects models provide coefficients of similar magnitude to the fixed effects models, with the exception of the model for the association between taxes and cigarette consumption in HICs in which a random-effects model was also reported. Three main panel regression models were fitted for each outcome with: i) total cigarette tax as a continuous term, which was further stratified by income level (HICs vs. LMICs); ii) total tax as a categorical variable (described above); iii) each type of tax as separate continuous variables in the same model. Model building was conducted through an iterative process which explored both linear and non-linear associations of exposure variables with the outcomes, considering model fit based on the Akaike and Bayesian Information Criteria (AIC and BIC). Results in the neonatal and infant mortality models should be interpreted as the relative change (as a percentage) in the outcome per 10 percentage-point increase in taxation as a percentage of the retail price of cigarettes because of log-transformation. Results in the cigarette consumption models should be interpreted as an absolute change in annual cigarette consumption per adult (number of sticks). Coefficients for rural population, drinking water, and clean cooking variables are presented per 10% change in their values, results for fertility and implementation of MPOWER tobacco control policies are presented per 1 unit change, results for total and female education are presented per 10 units change, and results for GDP per capita and health expenditure are presented per 1,000 PPP change.

We conducted sensitivity analyses without missing values estimated using linear interpolation, as well as without controlling for female primary education completion rate, as this was the covariate with the most missing values (8.1% after linear interpolation).

We applied effect estimates of total tax (as a continuous variable) on neonatal and infant mortality to the absolute number of deaths by country in 2018 (obtained from the World Bank [32]). The potential numbers of neonatal and infant deaths that could have been avoided in 2018 if countries had implemented higher levels of cigarette taxation were computed considering two scenarios: (1) if all countries increased their taxes by 10 percentage points or (2) if countries which did not adhere to WHO recommendations had increased taxation on cigarettes to 75% of the retail price. We calculated these absolute effects globally and by country-level income groups.

### Ethics approval

We used publicly available aggregated country-level data in this study and ethical approval was not required.

## Results

On average, the neonatal mortality rate was 14.4 and the infant mortality rate was 24.9 per 1,000 live births globally between 2008 and 2018 (S1 Table). The average neonatal and infant mortality rates were considerably higher in LMICs (19.0 and 33.2, respectively) than in HICs (3.7 and 5.6, respectively). Between 2008 and 2018, the average total tax on cigarettes was 49.1% with this figure lower in LMICs than in HICs (42.7% vs 63.8%) (S1 and S2 Tables). In 2018, only 20.4% of all countries– 11.2% of LMICs, and 42.1% of HICs–achieved the tax level recommended by WHO (i.e. ≥75% of the retail price).

Fig 1 shows the regression coefficients for changes in neonatal and infant mortality associated with changes in total cigarette tax and by type of tax (full results from analyses in S3 and S4 Tables). Both the continuous and categorical variables of total cigarette tax were inversely

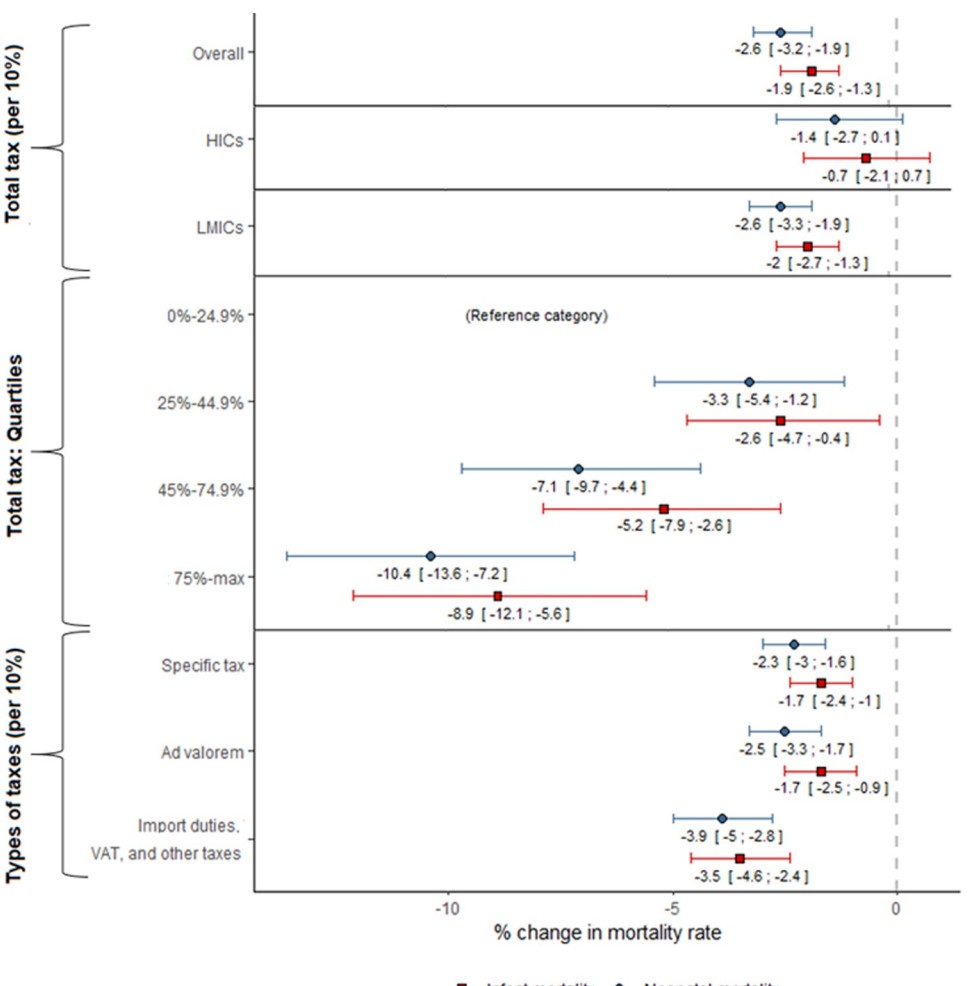

**Fig 1. Association between tobacco taxes and neonatal and infant mortality by different country-level income groups, tobacco tax quartiles, and the type of taxes (% change and 95% CI).** Note: This figure contains only the exposure variables; the entire analysis is available in S3 and S4 Tables (appendix pp 4–5). Tax is expressed as a percentage of the retail price. Abbreviations: CI = Confidence interval; VAT = value-added tax; HICs = High-income countries; LMICs = Low- and middle-income countries.

associated with neonatal and infant mortality. In fully adjusted models, a 10 percentage-point increase in total tax (relative to the retail price) was associated with a 2.6% (95% CI: 1.9% to 3.2%) decrease in neonatal mortality and a 1.9% (95% CI: 1.3% to 2.6%) decrease in infant mortality. In absolute terms, a global 10 percentage-point increase in total tax would have prevented 77,946 (95% CI: 49,555 to 106,130) infant deaths and 64,177 (95% CI: 46,570 to 81,653) neonatal deaths in 2018. Furthermore, if all countries had instituted at least 75% total tax as recommended by WHO, an estimated 231,220 (95% CI: 152,658 to 307,655) infant deaths and 181,970 (95% CI: 135,679 to 226,377) neonatal deaths, could have been averted in 2018 (Fig 2).

Differences in effect estimates were identified across country-level income groups (Fig 1 and S3 Table). In LMICs, a 10 percentage-point increase in total tax was associated with a 2.6% (95% CI: 1.9% to 3.3%) decrease in neonatal mortality and a 2.0% (95% CI: 1.3% to 2.7%) decrease in infant mortality. Estimates in HIC were in the same direction, however, confidence intervals were wide and crossed zero. Of the estimated 231,220 averted infant deaths in 2018, 99.2% were in LMICs (Fig 2).

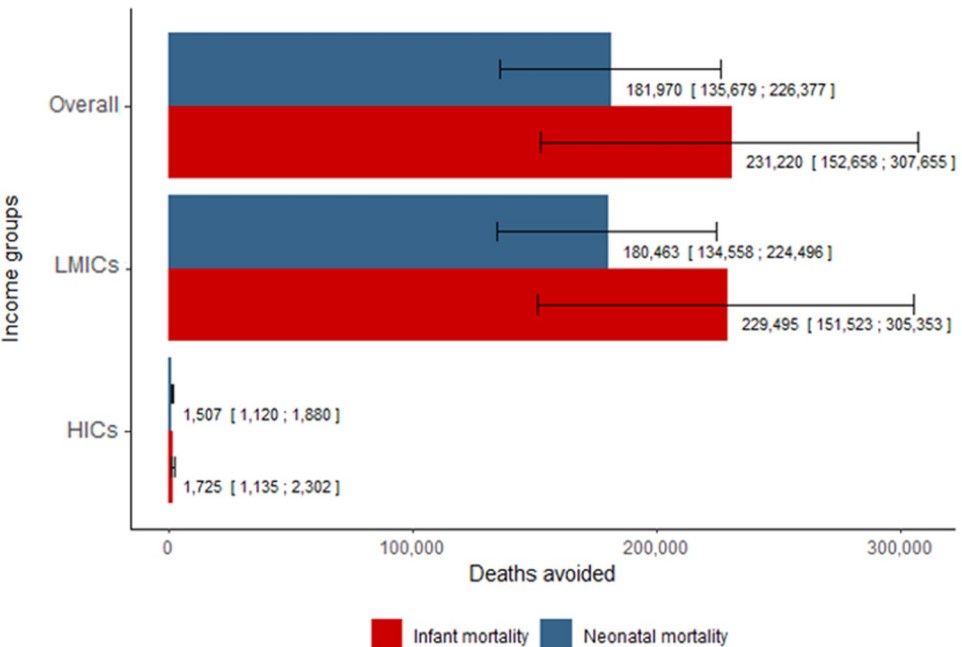

**Fig 2. Neonatal and infant deaths averted by raising taxes to 75% by income group (number of deaths and 95% CI).** Abbreviations: VAT = value-added tax; HICs = High-income countries; LMICs = Low- and middle-income countries.

A 10 percentage-point increase in specific tax and ad valorem tax was associated with a similar level of reduction in neonatal (2.3% vs 2.5%, respectively) and infant mortality (1.7% in both cases), whereas the same increase in import duties, VAT, or other taxes was associated with a larger decrease in both outcomes (3.9% in neonatal mortality and 3.5% in infant mortality) (Fig 1).

Higher total taxes were associated with lower cigarette consumption in the subset of countries where data were available (Fig 3, full regression results in S5 and S6 Tables). In fully adjusted fixed-effects models, a 10 percentage-point increase in total cigarette tax was associated with a –94.6 (95% CI: –156.5 to –32.7) per capita change in annual cigarette consumption. This change was –86.2 in LMICs (95% CI: –167.3 to –5.0) and –105.5 in HICs (95% CI:– 258.8 to 47.9). This association was significant for VAT, import duties, or other taxes (–678.4 [95% CI: –970.2 to –386.7]) and specific taxes (–88.5 [95% CI: –155.9 to –21.1]), but not for ad valorem taxes (–38.3 [95% CI: –105.1 to 28.5]). The more conservative random-effects model indicated that the association between total tax and cigarette consumption was statistically significant even in HICs (–84.1 [–155.2; –12.9]).

Findings were robust in sensitivity analyses which did not use linear interpolation for years without data collection (S7 Table), as were those excluding female primary education completion rate as a covariate (S8 Table) which allowed a higher number of observations (1895 observations from 175 countries compared with 1709 observations from 159 in the main model).

## Discussion

In this global analysis of 159 countries, higher cigarette taxes were associated with significant declines in both neonatal and infant mortality–with virtually all health gains in LMICs.

These findings are consistent with limited existing evidence showing cigarette taxation benefits infant health in HICs [10–13]. This analysis is the first to show that these benefits extend

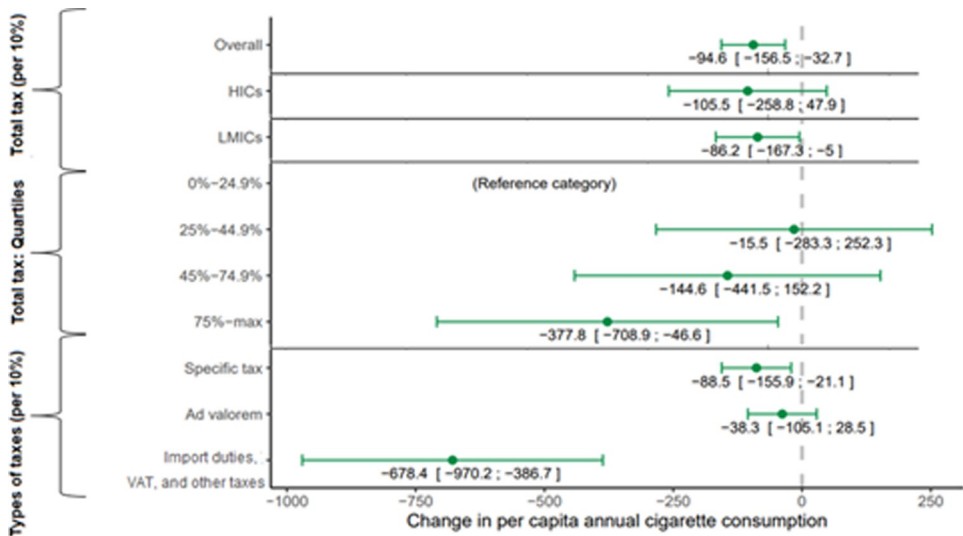

**Fig 3. Association between tobacco taxes and cigarette consumption per capita by different income groups, tobacco tax quartiles, and the type of taxes (B-value and 95% CI).** Note: This figure contains only the exposure variables; the entire analysis is available in S6 Table (appendix p 7). The B-values are all derived from a fixed-effect model. Tax is expressed as a percentage of the retail price. Abbreviations: CI = Confidence interval; VAT = value-added tax; HICs = High-income countries; LMICs = Low- and middle-income countries.

to countries globally and especially in LMICs which have higher burdens of child mortality and lower levels of cigarette taxation than HICs. The plausibility of our findings is supported by evidence which links increased taxation to higher prices, which in turn affect tobacco use and SHS [33]. Therefore, health impacts of taxation are likely mediated through decreases in prenatal and postnatal SHS exposure and smoking during pregnancy [5,34]. We found that cigarette consumption per capita indeed decreased significantly following increases in cigarette taxes, which lends support to causal interpretation of our findings.

We additionally investigated associations between different types of tobacco taxes and neonatal and infant mortality. We found that increases in all types of taxation were associated with benefits for child survival. The WHO recommends using specific taxes over ad valorem [35] and there is evidence that uniform tax structures based on specific taxes are the most effective in reducing consumption [36]. This recommendation is consistent with our findings on cigarette consumption where increases in ad valorem tax were not associated with a significant reduction. Although increasing VAT/import duties/other taxes seemed to have the largest benefit, this result needs to be viewed in light of the potential for raising each tax category. Tobacco excise taxes can be and have been increased substantially in many countries, whereas VAT and import duties generally affect a wide range of products and can rarely change more than a few percentage points. Furthermore, VAT increases may also reduce consumption of other harmful products, such as alcohol, which potentially adds to the effect on child survival.

This study used a range of comparable data and a quasi-experimental design; shown to be appropriate within this context and evidence indicates such approaches can obtain effect estimates similar to randomised controlled trials. There are nonetheless limitations which should be borne in mind when interpreting these findings. Although it was not possible to control for all potential confounders, our fixed-effects analyses controlled for both observed and unobserved time-invariant heterogeneity and we additionally controlled for the most important time varying variables. Data were available annually for mortality but only biennially for taxation and other tobacco control policies at the country-level. We addressed this by using linear

interpolation for intermediate years. This approach is justified on the basis that these changes are likely to be gradual over time, and allows a wider range of data to be used, although it may introduce some imprecision. Nonetheless, we conducted sensitivity analyses without these interpolated data which came to similar conclusions. In this analysis, we necessarily assumed that associations between taxation and mortality are similar across countries. This may not be entirely true, as we were unable to account for a number of factors which might influence the link between taxes and child survival, such as the lagged prevalence of smoking, the extent of illicit trade [37], tax, prices, and availability of other harmful tobacco and non-tobacco products, sales of single cigarettes that are more affordable for low-income groups [38] or health-care reforms targeting neonatal and infant mortality directly. Data on such measures and tactics are not systematically and comparably available for the wide range of countries studied here, meaning that our findings should only be applied to individual countries with caution. Also, while we adjusted models for country level MPOWER scores, these may not fully capture granular details of tobacco control measures at the local level. Finally, we have focused on cigarette taxes and not prices which more directly affect smoking behaviours. Tobacco control efforts to raise prices through taxation are often offset by tobacco companies using a range of strategies to circumvent increases [39,40]. Nevertheless, our analysis aims to inform policymakers who do not usually set prices directly, but rather influence them via taxation.

Current child mortality trends suggest that further efforts are needed globally to realise SDG 3.2. Our findings point to substantial gains in child survival from implementing WHO-recommended tobacco taxation especially in LMICs where 98.7% of global infant deaths take place [41]. Taxation is the least implemented of the WHO MPOWER measures, with only 14% of the world's population covered by recommended levels of tobacco taxation at the latest WHO assessment [8]. Our study provides crucial new evidence to strengthen advocacy for increased tobacco taxation to protect child health. Increased taxation alongside other tobacco control measures should be better integrated into national and global strategies to reduce neonatal and infant mortality to accelerate progress in the achievement of SDG targets. This is especially the case in LMICs where the delivery of health care interventions recommended in the 'Global Strategy for Women's, Children's and Adolescent's Health' to protect child health remain challenging. The revenue raised from tobacco taxes can be used to help finance these interventions alongside other health promotion interventions [42].

Future research could build on our findings by assessing the impacts of the other MPOWER measures across multiple countries. Previous evidence is limited by the fact that it is based on country or regional level case studies, limiting external validity [10,31,43,44]. While our current study did include information about other MPOWER policies as covariates, these should not be interpreted causally since the choice of models and covariates was driven by the need to adjust for potential confounders associated with taxation and child survival outcomes specifically. Further analyses could strengthen causal inference of the link between taxation and child survival by better exploring causal pathways, including the relative contributions of prenatal and postnatal SHS exposure. Finally, future research should further investigate the differences between countries in improving child survival by raising taxes since this link depends on a complex range of market forces, including the influence of the tobacco industry.

## Conclusion

Raising cigarette taxes is associated with substantial reductions in neonatal and infant mortality globally, particularly in LMICs. This finding reinforces the pressing need for countries to implement WHO recommended levels of tobacco taxation as an integral component of their national strategies to achieve SDG 3.2.

## Supporting information

**S1 Table. Summary statistics of the included variables (2008–2018).** Abbreviations:
VAT = value-added tax; GDP = Gross domestic product; PPP = Purchasing power parity.
(DOCX)

**S2 Table. Summary statistics of the included variables according to income groups (2008–2018).** Abbreviations: VAT = value-added tax; GDP = Gross domestic product;
PPP = Purchasing power parity.
(DOCX)

**S3 Table. Results from the fixed effects panel regression model for the association between total taxes and neonatal and infant mortality (Ratios and 95% Confidence Interval).** Note:
We reported ratios (i.e. exponential values of effect estimates) from regression models with
log-transformed neonatal and infant mortality outcomes. Hausman Test indicated for each
model that fixed effect model is the preferred model. Abbreviations: VAT = value-added tax;
GDP = Gross domestic product; PPP = Purchasing power parity, AIC = Akaike information
criterion; BIC = Bayesian information criterion.
(DOCX)

**S4 Table. Results from the fixed effects panel regression model for the association between different types of taxes and neonatal and infant mortality (Ratios and 95% Confidence Interval).** Note: We reported ratios (i.e. exponential values of effect estimates) from regression
models with log-transformed neonatal and infant mortality outcomes. Hausman Test indicated for each model that fixed effect model is the preferred model. Abbreviations:
VAT = value-added tax; GDP = Gross domestic product; PPP = Purchasing power parity,
AIC = Akaike information criterion; BIC = Bayesian information criterion.
(DOCX)

**S5 Table. Results from the fixed-effects panel regression model for the association between taxes and cigarette consumption without any control variables (B-value and 95% Confidence Interval).** Note: Hausman Test indicated for each model that fixed effect model is the
preferred model, except in case of the model for high-income countries (in this case the preferred model is the random effect model that would give the following B-values [95% Confidence interval]: -119.1 [-198.5; -49.6]). Abbreviations: VAT = value-added tax; GDP = Gross
domestic product; PPP = Purchasing power parity, AIC = Akaike information criterion;
BIC = Bayesian information criterion.
(DOCX)

**S6 Table. Results from the fixed-effects panel regression model for the association between taxes and cigarette consumption (B-value and 95% Confidence Interval).** Note: Hausman
Test indicated for each model that fixed effect model is the preferred model, except in case of
the model for high-income countries (in this case the preferred model is the random effect
model that would give the following B-values [95% Confidence interval]: -84.1 [-155.2; -12.9]).
Abbreviations: VAT = value-added tax; GDP = Gross domestic product; PPP = Purchasing
power parity, AIC = Akaike information criterion; BIC = Bayesian information criterion.
(DOCX)

**S7 Table. Results from the fixed-effects panel regression model for neonatal and infant mortality (Ratios and 95% Confidence Interval) and cigarette consumption (B-value and 95% Confidence Interval) without missing value imputation.** Note: We reported ratios (i.e.
exponential values of effect estimates) from regression models with log-transformed neonatal

and infant mortality outcomes. Hausman Test indicated for each model that fixed effect model is the preferred model. Abbreviations: VAT = value-added tax; GDP = Gross domestic product; PPP = Purchasing power parity, AIC = Akaike information criterion; BIC = Bayesian information criterion.
(DOCX)

**S8 Table. Results from the fixed effects panel regression model for the association between total taxes and neonatal and infant mortality excluding education variables (Ratios and 95% Confidence Interval).** Note: We reported ratios (i.e. exponential values of effect estimates) from regression models with log-transformed neonatal and infant mortality outcomes. Hausman Test indicated for each model that fixed effect model is the preferred model. Abbreviations: VAT = value-added tax; GDP = Gross domestic product; PPP = Purchasing power parity, AIC = Akaike information criterion; BIC = Bayesian information criterion Akaike information criterion; BIC = Bayesian information criterion.
(DOCX)

**S1 Text.**
(DOCX)

## Author Contributions

**Conceptualization:** Márta K. Radó, Anthony A. Laverty, Thomas Hone, Kiara Chang, Mohammed Jawad, Christopher Millett, Jasper V. Been, Filippos T. Filippidis.

**Data curation:** Márta K. Radó, Filippos T. Filippidis.

**Formal analysis:** Márta K. Radó, Anthony A. Laverty, Christopher Millett, Jasper V. Been, Filippos T. Filippidis.

**Investigation:** Márta K. Radó, Anthony A. Laverty, Filippos T. Filippidis.

**Methodology:** Jasper V. Been, Filippos T. Filippidis.

**Supervision:** Anthony A. Laverty, Thomas Hone, Christopher Millett, Jasper V. Been, Filippos T. Filippidis.

**Visualization:** Márta K. Radó.

**Writing – original draft:** Márta K. Radó, Anthony A. Laverty, Filippos T. Filippidis.

**Writing – review & editing:** Thomas Hone, Kiara Chang, Mohammed Jawad, Christopher Millett, Jasper V. Been.

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
