## [Decision Letter · Decision Letter 0]

23 Sep 2021

 PGPH-D-21-00545

 Cigarette taxation and neonatal and infant mortality: a longitudinal analysis of 159 countries

Dear Dr. Radó,

Thank you for submitting your manuscript to PLOS Global Public Health. After careful consideration, we feel that it has merit but does not fully meet PLOS Global Public Health’s publication criteria as it currently stands. Therefore, we invite you to submit a revised version of the manuscript that addresses the points raised during the review process.

We look forward to receiving your revised manuscript.

Kind regards,

Ransome Eke, M.D., Ph.D., MPH

Academic Editor

Journal Requirements:

1. Please amend your detailed Financial Disclosure statement. This is published with the article, therefore should be completed in full sentences and contain the exact wording you wish to be published.

2. Please note that your Data Availability Statement is currently missing the repository name and a direct link to access each database. If your manuscript is accepted for publication, you will be asked to provide these details on a very short timeline. We therefore suggest that you provide this information now, though we will not hold up the peer review process if you are unable.

3. Please provide separate figure files in .tif or .eps format only and remove any figures embedded in your manuscript file. Please ensure that all files are under our size limit of 20MB.  

Once you've converted your files to .tif or .eps, please also make sure that your figures meet our format requirements.

4. We have noticed that you have uploaded supporting information but you have not included a list of legends.  Please add a full list of legends for all supporting information files (including figures, table and data files) after the references list.

Additional Editor Comments (if provided):

Please address the comments from reviewer 2

Reviewers' comments:

Reviewer's Responses to Questions

**Comments to the Author**

1. Does this manuscript meet PLOS Global Public Health’s publication criteria? Is the manuscript technically sound, and do the data support the conclusions? The manuscript must describe methodologically and ethically rigorous research with conclusions that are appropriately drawn based on the data presented.

Reviewer #1: Yes

Reviewer #2: Yes

2. Has the statistical analysis been performed appropriately and rigorously?

Reviewer #1: Yes

Reviewer #2: Yes

3. Have the authors made all data underlying the findings in their manuscript fully available (please refer to the Data Availability Statement at the start of the manuscript PDF file)?

Reviewer #1: Yes

Reviewer #2: Yes

4. Is the manuscript presented in an intelligible fashion and written in standard English?

Reviewer #1: Yes

Reviewer #2: Yes

5. Review Comments to the Author

Reviewer #1: The manuscript has tried to assess the impact of cigaratte taxation against neonatal/infant mortality, of course there are many confounding factors. This information could serve as an evidence for the advocacy towards smoking ban and or smoking free legislations...

It would have been better if they included smoking free legislations in the objectives of their work

In the materials and methods, construction of time-series models by infant race for cigarette tax and price with infant mortality would have been better to be included as the outcome.

If possible, the authors are encouraged to think of the impact of educational attainments of the smokers (if possible), mean inflation-adjusted per-capita income of the countries, and country random effects; so that it is possible to appreciate the role of cigarette tax on the deaths

The authors are expected to explain confident they are to conclude the cause of death is exclusively because of cigarettes; OR the reduction of death is exclusively because of taxation of cigarette? Because there are several reasons to mention. So it is better to bring justification for the confounding factors

I didn’t see results for specific excise, ad valorem, Value Added Tax (VAT)/ import duties/ other taxes separately and which of the types of the taxes had greater impact on the deaths?

Did u see any disparity towards cigarette tax and infant/neonatal death among countries??

I couldn’t find any of the figures in this text

Any hint for which countries are included for this specific study?

Confounding factors are there to be discussed

There are countries with smoking free legislations and how did you see this issue?

My other concern is that whatever legislation and smoking free legislation you impose, people who are addicted for cigarettes will continue to smoke……..

In the recommendation part, What if you recommend smoking bans?

Reviewer #2: Abstract:

1. write the aim of the study after the introduction

2. abstract includes conclusions and recommendations not a discussion

introduction

line 67 the referernce is a systematic review saying the studies on taxation on child health is fewer. I think this should be revised

Methods:

1- The study design: it is not a quasi-epxerimental study. there is no intervention applied by the researchers. It is more likely a secondary data analysis

2- table 1 i think it could be a supplementary

Results:

1- when saying 2.6% reduction or decreas, no need for the minus (-). It is already understandable by the word decrease

2- all tables are supplementary, i think the researcher can choose one or two to put them in the results

6. PLOS authors have the option to publish the peer review history of their article (what does this mean?). If published, this will include your full peer review and any attached files.

**Do you want your identity to be public for this peer review?** For information about this choice, including consent withdrawal, please see our Privacy Policy.

Reviewer #1: **Yes: **Dr Fufa Abunna Kurra

Reviewer #2: No

---

## [Decision Letter · Decision Letter 1]

15 Nov 2021

Cigarette taxation and neonatal and infant mortality: a longitudinal analysis of 159 countries

PGPH-D-21-00545R1

Dear Dr. Radó,

We're pleased to inform you that your manuscript has been judged scientifically suitable for publication and will be formally accepted for publication once it meets all outstanding technical requirements.

Within one week, you'll receive an e-mail detailing the required amendments. When these have been addressed, you'll receive a formal acceptance letter and your manuscript will be scheduled for publication.

An invoice for payment will follow shortly after the formal acceptance. To ensure an efficient process, please log into Editorial Manager at https://www.editorialmanager.com/pgph/ click the 'Update My Information' link at the top of the page, and double check that your user information is up-to-date. If you have any billing related questions, please contact our Author Billing department directly at authorbilling@plos.org.

Kind regards,

Ransome Eke, M.D., Ph.D., MPH

Academic Editor

Additional Editor Comments (optional):

Reviewers' comments:

Reviewer's Responses to Questions

**Comments to the Author**

1. If the authors have adequately addressed your comments raised in a previous round of review and you feel that this manuscript is now acceptable for publication, you may indicate that here to bypass the “Comments to the Author” section, enter your conflict of interest statement in the “Confidential to Editor” section, and submit your "Accept" recommendation.

Reviewer #1: All comments have been addressed

Reviewer #2: All comments have been addressed

2. Does this manuscript meet PLOS Global Public Health’s publication criteria? Is the manuscript technically sound, and do the data support the conclusions? The manuscript must describe methodologically and ethically rigorous research with conclusions that are appropriately drawn based on the data presented.

Reviewer #1: Yes

Reviewer #2: Yes

3. Has the statistical analysis been performed appropriately and rigorously?

Reviewer #1: Yes

Reviewer #2: Yes

4. Have the authors made all data underlying the findings in their manuscript fully available (please refer to the Data Availability Statement at the start of the manuscript PDF file)?

Reviewer #1: Yes

Reviewer #2: Yes

5. Is the manuscript presented in an intelligible fashion and written in standard English?

Reviewer #1: Yes

Reviewer #2: Yes

6. Review Comments to the Author

Reviewer #1: (No Response)

Reviewer #2: (No Response)

7. PLOS authors have the option to publish the peer review history of their article (what does this mean?). If published, this will include your full peer review and any attached files.

**Do you want your identity to be public for this peer review?** For information about this choice, including consent withdrawal, please see our Privacy Policy.

Reviewer #1: **Yes: **Dr Fufa Abunna Kurra

Reviewer #2: No
